

# French validation of the Barcelona Music Reward Questionnaire

Joe Saliba[1], Urbano Lorenzo-Seva[2], Josep Marco-Pallares[3,4], Barbara Tillmann[5], Anthony Zeitouni[1] and Alexandre Lehmann[1,6]

[1] Department of Otolaryngology—Head and Neck Surgery, McGill University, Montreal, Canada
[2] Research Center for Behavior Assessment, Universitat Rovira I Virgili Tarragona, Tarragona, Spain
[3] Department of Basic Psychology, Universitat de Barcelona, Barcelona, Spain
[4] Cognition and Brain Plasticity Group, Institut d'Investigacions Biomèdiques de Bellvitge (IDIBELL), L'Hospitalet de Llobregat, Spain
[5] Team Auditory Cognition and Psychoacoustics, Lyon Neurosciences Research Center, CNRS-UMR 5292, INSERM U1028, University Lyon 1, Lyon, France
[6] International laboratory for Brain, Music and Sound Research (BRAMS), Center for Research on Brain, Language and Music (CRBLM), Montreal, Canada

Corresponding author
Joe Saliba, joe.saliba@mail.mcgill.ca

## ABSTRACT

**Background.** The Barcelona Music Reward Questionnaire (BMRQ) questionnaire investigates the main facets of music experience that could explain the variance observed in how people experience reward associated with music. Currently, only English and Spanish versions of this questionnaire are available. The objective of this study is to validate a French version of the BMRQ.

**Methods.** The original BMRQ was translated and adapted into an international French version. The questionnaire was then administered through an online survey aimed at adults aged over 18 years who were fluent in French. Statistical analyses were performed and compared to the original English and Spanish version for validation purposes.

**Results.** A total of 1,027 participants completed the questionnaire. Most responses were obtained from France (89.4%). Analyses revealed that congruence values between the rotated loading matrix and the ideal loading matrix ranged between 0.88 and 0.96. Factor reliabilities of subscales (i.e., Musical Seeking, Emotion Evocation, Mood Regulation, Social Reward and Sensory-Motor) also ranged between 0.88 and 0.96. In addition, reliability of the overall factor score (i.e., Music reward) was 0.91. Finally, the internal consistency of the overall scale was 0.85. The factorial structure obtained in the French translation was similar to that of the original Spanish and English samples.

**Conclusion.** The French version of the BMRQ appears valid and reliable. Potential applications of the BMRQ include its use as a valuable tool in music reward and emotion research, whether in healthy individuals or in patients suffering from a wide variety of cognitive, neurologic and auditory disorders.

## INTRODUCTION

The rewarding effects of music are highly dependent on cultural and personal preferences. As a result, large differences in the way individuals experience musical pleasure are observed (*Blood & Zatorre, 2001*; *Chanda & Levitin, 2013*). While music can induce positive

effects on mood and affect in some individuals (*Juslin & Västfjäll, 2008*), others seek the social bonding capacity of music (*Cross, 2001*). Conversely, certain individuals cannot experience any pleasure from widely different stimuli, including music—a disorder termed anhedonia (*Assogna et al., 2011*). Traditionally, it has been hard to assess the sources of this inter-individual variability in music-induced reward. Previous groups have developed questionnaires—such as the BIS/BAS scales (*Carver & White, 1994*) or the Sensitivity to Reward/Sensitivity to Punishment Questionnaire (*Torrubia et al., 2001*)—that assess individual differences to overall sensitivity to reward experiences (*Carver & White, 1994*; *Torrubia et al., 2001*). However, music is considered as a higher-order pleasure and, as such, might involve different processing mechanisms than basic rewards (*Menon & Levitin, 2005*). In addition, previous studies have supported a dissociation of music rewarding experience from other rewarding experiences related to other types of primary and secondary reinforcements, such as food, sex and money among others (*Mas-Herrero et al., 2014*).

In light of these findings, the Barcelona Music Reward Questionnaire (BMRQ) was developed by *Mas-Herrero et al. (2013)*. This questionnaire is specifically geared towards assessing sensitivity to music reward and was a welcome addition to a limited choice of behavioral tools suitable for music reward studies. The BMRQ can serve as a valuable research tool in psychophysical studies addressing music reward in healthy individuals, hearing-impaired individuals or individuals affected with other conditions or pathologies. To date, only Spanish and English versions of this questionnaire are available in the literature, limiting its application. With over 200 million speakers worldwide, French is one of the most common languages in the world. In countries with several official languages including French such as Canada, Morocco and Senegal, it is all the more important for research tools to be available in all official languages to adequately test the population. In that context, a French version of the BMRQ was required to meet the needs of the numerous researchers in the French-speaking areas of the world. In this paper, we sought to translate the Barcelona Music Reward Questionnaire into an international French and to assess its construct validity and reliability.

## MATERIALS AND METHODS

### The Barcelona Music Reward Questionnaire

The BMRQ examines five main facets that characterize musical reward experience in individuals: musical seeking, emotion evocation, mood regulation, social reward and sensory-motor. Musical seeking refers to the way individuals pursue music-related activities (attending concerts, playing an instrument) or seek additional information about music they listen to (performers, composers). The emotion evocation aspect of music reward is related to the emotional impact of music on individuals. In contrast, the ability of listeners to use music to modulate their emotions (to relieve stress, to release emotions, to comfort) is referred to as mood regulation. The social reward facet examines the social bonding effect of music on individuals. Lastly, the sensory-motor facet addresses the capacity of music to intuitively induce body movements synchronized to a rhythm's beat in certain individuals (head nodding, even dancing). The questionnaire contains 20 statements

equally divided among these five facets. Participants indicate the level of agreement with each statement by using a five-point scale ranging from "fully disagree" (1) to "fully agree" (5). The contribution of each facet to the overall music reward experience is quantified by a numerical value obtained upon completion of the survey. A score for global sensitivity to music reward is also provided, which was obtained as the weighted sum of participants' scores (i.e., factor score). The mean value of each factor is 50, and the standard deviation is 10. Standard values are therefore located between 40 and 60. Punctuations below 40 indicate low values in this particular facet, whereas values above 60 indicate high values (the same applies to the global sensitivity to music reward) (*Mas-Herrero et al., 2013*).

The BMRQ was created in three steps (*Mas-Herrero et al., 2013*). The first consisted in developing of a short psychometric instrument in Spanish that included various facets of music and reward experiences. This initial instrument included 112 items addressing a variety of activities and situations associated with music reward and pleasure experiences, and was administered to 804 Spanish participants. From the initial pool of 112 items, only 20 were retained for the final version, equally divided among five facets of music reward (music seeking activities, mood regulation, emotion evocation, sensory-motor behavior and social reward). Selection was based on loading values and content and adequacy of the items. The second step involved exploratory and confirmatory factorial analysis of the Spanish BMRQ. The questionnaire with the selected 20 items was administered to a new sample of 605 students in an effort to replicate the previous findings. Analyses revealed a reliable factorial structure for the Spanish BMRQ and an acceptable fit for the hypothesized five facets of music reward. The final step in the development of the BMRQ was its translation and adaptation into English. The translated version was completed by 252 English-speaking participants, and confirmatory factorial analysis was performed to verify the replicability of the factor structure obtained in the Spanish version. The original BMRQ has been shown to be valid with acceptable reliability estimates of factor scores (i.e., a reliability value of 0.93 for the overall scale, and values between 0.73 and 0.93 for the five subscales). The aim of the present study was to replicate this validation for a French adaptation of the BMRQ.

## Questionnaire translation

The French adaptation of the BMRQ was obtained by forward and backward translation. Each item in the original English version of the questionnaire was independently translated by two groups two of bilingual (French and English) researchers—in Montreal, Canada and in Lyon, France—whose first language was French. Both groups also had knowledge of the subject matter. The groups were purposely chosen in different geographic areas in order to account for the regional differences in spoken French and hence create an internationally comprehensible French translation. The Spanish questionnaire was used as a reference for disambiguating some wordings. The emphasis was placed on the translation of meaning rather than a literal one. A consensus between the two translator groups was obtained to produce the final French version of the questionnaire. Finally, a third bilingual researcher (French and Spanish) conducted a back translation into Spanish. This researcher was not involved in the initial translation process. This last step ensured the meaning of the adapted
French version was concordant with the meaning of the original Spanish questionnaire. The comparison between the source items and the French translation is shown in Appendix S1. The content of the French translation of the BMRQ is reproduced in Appendix S2, along with the complete set of instructions, thus allowing readers for a direct use of this questionnaire tool.

## Data collection and participants

The questionnaire was administered via an Internet platform (LimeSurvey, McGill University servers) to any participant aged over 18 years and fluent in French. A written electronic consent was obtained for each participant. The survey was made publicly accessible from November 2014 to April 2015 and distributed electronically through various academic and healthcare institutions mainly in Europe, North America and Africa, but also in other areas of the world. In order to avoid sampling bias effect, the music focus of the study was not explicitly stated in the test instructions when administered to participants. Prior to completing the survey, participants were also asked to fill out a general demographic and linguistic background questionnaire. This study was approved by the McGill University's Faculty of Medicine Institutional Review Board (#A11-E88-14B).

## Evaluation of the psychometric properties of the translated version

In order to assess the structure validity of the test, we used an approach similar to that employed by *Mas-Herrero et al. (2013)* in the development of the original questionnaire, as described above. An exploratory factor analysis was carried out using MATLAB, and, for scale analyses, SPSS 22 was used. The polychoric correlation matrix was computed for the 20 items of the translated questionnaire. To control the variance due to this response style factor, we applied the procedure proposed by *Lorenzo-Seva & Rodriguez-Fornells (2006)* developed for the specific case of non-perfectly balanced scales (see *Lorenzo-Seva & Ferrando, 2009*). As five content factors were expected, we retained this number of factors using Minimum Rank Factor Analysis (MRFA, *Ten Berge & Kiers, 1991*). Observed variables in MRFA consist of common parts and unique parts, each satisfying certain requirements: the covariance matrices for common and unique parts are positive semidefinite, and the unique-parts covariance matrix is diagonal. An oblique semi-specified Procrustean rotation (*Browne, 1972*) was performed in order to establish the loading factors associated with each of the five content factors. For the purposes of this analysis, the specified values were the loadings on each item that we expected to be zero. The procedure reported by *Ten Berge et al. (1999)* was then employed to calculate factor scores. The mean and standardized deviation of items, and the factor weights required to compute these factor scores are available for the use of researchers (in the Supplemental Information and on the online test page).

We computed the reliability estimates for the five scales and the total scale on the basis of the factor scores based on the factor scores reliability (for example, see *Mellenbergh, 1994*; formula 22 on page 231). To assess internal consistency, we computed Cronbach's alpha for the overall scale.
**Table 1** Demographics.

| Variable | Participants ($n = 1,027$) |
|---|---|
| **Age, years** | |
| Mean (SD) | 22.3 (7.8) |
| Minimun | 18 |
| Maximum | 54 |
| **Gender, *n* (%)** | |
| Male | 363 (35.3) |
| Female | 664 (64.7) |
| **Education, *n* (%)** | |
| University | 788 (76.7) |
| College/Professional degree | 218 (21.2) |
| High school | 21 (2.0) |
| **Country where French was learned, *n* (%)** | |
| France | 918 (89.4) |
| Canada | 52 (5.1) |
| Algeria | 10 (1.0) |
| Madagascar | 7 (0.7) |
| Belgium | 2 (0.2) |
| Other (25 countries) | 38 (3.7) |
| **Musician, *n* (%)** | |
| Yes | 231 (22.5) |
| No | 796 (77.5) |

## RESULTS

A total of 1,027 participants voluntarily completed the entire translated questionnaire (Mean age: 22.3 (SD 7.8) years, females: 64.7%). While participants were mostly from France (89.4%) and Canada (5.1%), 4% of our sample was obtained from 25 other countries such as Cameroun, Senegal and Egypt. Table 1 resumes the demographic statistics of the sample. The majority of our respondents were non-musicians (77.5%). While the questionnaire was primarily advertised in academic institutions, approximately a quarter of our participants did not complete a university degree. Overall, our French sample was similar to the Spanish and English samples in terms of age, gender and music training.

Table 2 shows the means and standard deviations of items of the French and the Spanish version of the test. As can be observed, the differences observed between the mean items in both cultures were not significant.

Once the polychoric correlation matrix was available, the observed Kaiser–Meyer–Olkin (KMO, *Kaiser, 1970*) index was computed: the 0.855 value obtained suggested that the correlation matrix was well suited for factor analysis (see *Kaiser & Rice, 1974*). The congruence values (*Tucker, 1951*) between the rotated loading matrix and the *ideal* loading matrix ranged from 0.88 to 0.96. As the coefficients were all above the threshold of 0.85,
**Table 2** Item–item comparison between the original Spanish scale and the adapted French version.

| Item | Original Mean (SD) | French Mean (SD) |
| --- | --- | --- |
| Q1 | 3.85 (0.86) | 4.00 (0.78) |
| Q2 | 1.70 (0.92) | 1.77 (1.01) |
| Q3 | 4.30 (0.79) | 4.32 (0.78) |
| Q4 | 4.17 (0.91) | 4.32 (0.88) |
| Q5 | 1.65 (1.02) | 2.11 (1.29) |
| Q6 | 3.74 (0.86) | 3.63 (0.99) |
| Q7 | 3.89 (0.98) | 3.72 (1.01) |
| Q8 | 4.53 (0.70) | 4.49 (0.76) |
| Q9 | 4.26 (0.82) | 4.33 (0.77) |
| Q10 | 3.96 (1.03) | 3.61 (1.24) |
| Q11 | 3.46 (0.86) | 3.49 (1.09) |
| Q12 | 3.55 (1.06) | 3.50 (1.36) |
| Q13 | 3.28 (1.25) | 3.12 (1.36) |
| Q14 | 4.35 (0.78) | 4.29 (0.85) |
| Q15 | 4.29 (0.76) | 4.21 (0.93) |
| Q16 | 3.82 (0.92) | 3.69 (0.99) |
| Q17 | 2.29 (1.12) | 2.17 (1.13) |
| Q18 | 3.94 (0.88) | 4.08 (1.01) |
| Q19 | 4.11 (0.98) | 4.12 (0.84) |
| Q20 | 4.00 (0.91) | 4.21 (0.90) |

**Notes.**
SD, standard deviation.

the factor similarity between the rotated loading matrix and the ideal loading matrix was fair (*Lorenzo-Seva & Ten Berge, 2006*). Table 3 shows not only the loading values after rotation, but also the loadings of items on the control scale (i.e., the acquiescence, AC). The procedure used to obtain a control scale was previously proposed by *Lorenzo-Seva & Ferrando (2009)*. The method is based on the idea that in a scale where some items are worded in the opposite direction to the other items, it is possible to identify acquiescent response style. In such a balanced scale, the centroid helps to estimate the overall tendency of individuals to use systematically a particular value of the response scales independently of the worded direction of the items (i.e., to show an acquiescent response style). In an initial step, the first centroid is computed, and it is taken as an estimate of the loading values of items on an underlying acquiescent factor. If the scale is partially balanced, a subset of balanced items is used to compute the first centroid, and then the unbalanced set of items is projected on the first centroid. The variance explained by the first centroid is then removed from the correlation matrix, and the residual correlation matrix is factor analyzed in order to estimate the loading on the content factors. In addition, this first centroid can be understood as a control scale: a scale that accounts for the variance due to the acquiescent response. Our results show that some of the items properly loaded on the AC scale. This finding confirms the appropriate use of a model in which AC response bias

**Table 3 Factorial loading matrix for each item of the adapted French version of the questionnaire.** Salient loading values (i.e., values larger than absolute 0.4) in the content factors are printed in bold face.

| Item | Acquiescence | Music seeking | Emotion evocation | Mood regulation | Sensory-motor | Social |
|------|------|------|------|------|------|------|
| Q11 | 0.047 | **0.790** | −0.060 | 0.108 | 0.108 | −0.056 |
| Q2 | 0.473 | **−0.629** | 0.019 | −0.290 | 0.005 | −0.037 |
| Q7 | 0.557 | **0.625** | 0.005 | 0.087 | −0.047 | 0.099 |
| Q17 | 0.193 | **0.605** | 0.126 | −0.173 | −0.077 | 0.270 |
| Q12 | −0.039 | 0.004 | **0.904** | −0.122 | 0.083 | −0.080 |
| Q8 | 0.057 | 0.056 | **0.856** | 0.021 | −0.002 | −0.001 |
| Q18 | 0.098 | 0.031 | **0.686** | 0.093 | −0.057 | 0.100 |
| Q3 | −0.055 | −0.117 | **0.634** | 0.208 | −0.041 | 0.059 |
| Q14 | −0.052 | 0.034 | −0.042 | **0.748** | 0.066 | 0.134 |
| Q9 | 0.007 | 0.218 | 0.101 | **0.680** | 0.057 | −0.134 |
| Q4 | −0.185 | 0.056 | 0.072 | **0.665** | 0.032 | −0.022 |
| Q19 | −0.023 | 0.131 | 0.241 | **0.641** | 0.046 | −0.011 |
| Q10 | 0.050 | 0.059 | −0.020 | −0.050 | **0.975** | −0.106 |
| Q5 | 0.236 | −0.055 | −0.005 | 0.253 | **−0.933** | 0.044 |
| Q20 | −0.013 | −0.041 | 0.013 | 0.314 | **0.527** | 0.158 |
| Q15 | −0.005 | −0.247 | 0.002 | 0.363 | **0.443** | 0.311 |
| Q1 | 0.018 | 0.033 | 0.039 | −0.044 | −0.052 | **0.705** |
| Q6 | 0.085 | 0.225 | −0.121 | 0.014 | 0.020 | **0.704** |
| Q13 | 0.082 | −0.039 | 0.114 | −0.013 | 0.144 | **0.591** |
| Q16 | 0.091 | 0.124 | 0.138 | −0.115 | 0.150 | **0.526** |

style was controlled. We were also able to validate that the loadings of items on the content factors were free of AC using this model. Lastly, using the loading values on the content factor, we demonstrated that the items in our adapted instrument were well related with the corresponding expected scale.

In addition, the inter-factor correlation values between content factors ranged between 0.22 and 0.32. While these inter-factor correlations are in general slightly lower than the original version of the test by Mas-Herrero (0.22–0.46), our results demonstrated that the scales were also correlated in the French adaptation.

Finally, the reliability estimates computed on the basis of the factor scores of the scales were 0.93, 0.96, 0.88, 0.91, and 0.93 for Musical Seeking, Emotion Evocation, Mood Regulation, Social Reward and Sensory-Motor, respectively. None of the reliability estimates obtained in our analyses were below the threshold of 0.80. In comparison, the corresponding reliability estimates in the original pooled English and Spanish samples were 0.89, 0.88, 0.87, 0.78, and 0.93, respectively (*Mas-Herrero et al., 2013*). Furthermore, the overall test (Music reward) in the French translation showed an acceptable reliability (0.91), concordant with the reported value by Mas-Herrero et al. (0.92). The distribution of the overall test scores (global sensitivity to music reward) using the French translation was centered on a mean of 50, similar to that of the original instrument (Fig. 1). Likewise, the internal consistency for the overall French scale was 0.852, with a 95% confidence interval

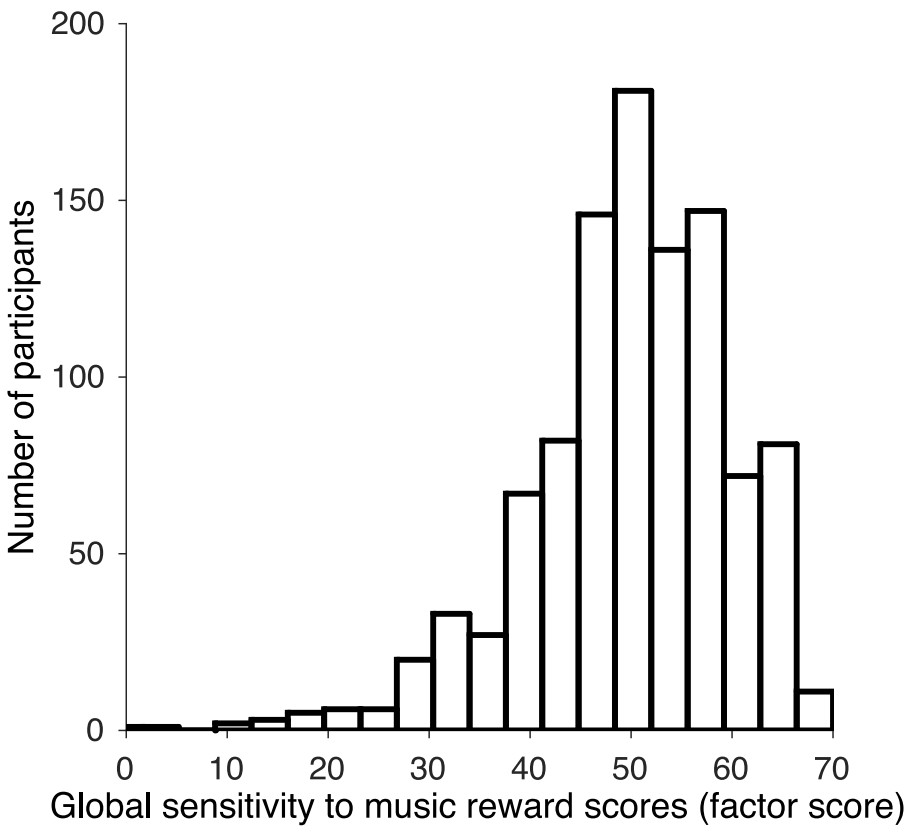

**Figure 1** Global sensitivity to music reward scores using the French version of the Barcelona Music Reward Questionnaire.

[0.839–0.865]. Globally, all our analyses demonstrated that the fit obtained in the French translation was similar to that of the original English and Spanish samples, indicating that the factorial structures are equivalent.

## DISCUSSION

Our study described the translation and adaptation of the BMRQ into French and provided analyses of the psychometric properties of the translated scale. Our results demonstrated that the translated BMRQ has acceptable construct validity while keeping the factorial structure of the original English and Spanish questionnaires. In general, the results that we obtained were similar to those reported by the developers of the original instrument (*Mas-Herrero et al., 2013*). This suggests that our translation procedure was successful.

The geographic distribution of French speakers encompasses over 30 countries throughout all five continents (*L'observatoire de la langue française, 2014*). With such a diverse speaker population, significant regional differences in the spoken language currently exist. In that context, an internationally acceptable French adaptation of the BMRQ was required to accommodate researchers and clinicians across the French-speaking regions. This translated BMRQ is born from a collaborative work between two bilingual groups in North America (Montreal, Quebec) and Europe (Lyon, France). Efforts were made

during the translation process to remove all regional French influences. Each group first independently translated the original English BMRQ into a locally acceptable French. Then, a consensus between the two translators was obtained to produce the final international French version of the questionnaire. We believe this collaboration was necessary to adapt the original BMRQ into a French that would be easily understood by speakers around the French-speaking world. This belief is echoed in our results: over 30 French-speaking countries are represented, and 5.5% of participants learned French in countries other than Canada (Quebec) or France. Finally, our association with the developers of the original Spanish instrument (UL) ensured the French adaptation remained faithful to the initial questionnaire.

While we collected responses from over 1,000 participants, the majority were obtained from France (89.4%). This is partly a reflection of the differences in the number of French speakers between the regions sampled: 6 million in Quebec compared to more than 77 million in the European Union (*L'observatoire de la langue française, 2014*). In an effort to reduce sampling bias effect, the music focus of the study was not explicitly stated in the test instructions when administered to participants. This can be seen in the number of non-musicians among our participants (77.5%), a proportion that is similar to what has been reported in the original version of the BMRQ. We therefore believe our sample is representative of the general French speaking population and that sampling bias was not significant.

Previous work by *Ayotte, Peretz & Hyde (2002)* and *Peretz, Champod & Hyde (2003)* have established that approximately 4% of the population suffers from congenital amusia, a disorder of music processing that hinders their ability to perceive, produce and enjoy music. In contrast, some individuals suffer from general anhedonia, a deficit in experiencing pleasure from widely different stimuli, usually in the context of depressive disorders or neurodegenerative diseases such as Parkinson's (*Loas et al., 1994*; *Assogna et al., 2011*). Three case studies have also reported a form of acquired anhedonia specific to music that resulted from strokes in limbic structures such as the amygdala, as well as areas of the temporo-parietal cortex, inferior parietal cortex and insula (*Mazzoni et al., 1993*; *Satoh et al., 2011*; *Griffiths et al., 2004*). In those neurologic and psychiatric patients, the use of a standardized tool such as the BMRQ will help determine a loss in the capacity of feeling emotions through music. However, the BMRQ can also be employed to explore music reward in healthy individuals. In fact, a recent report by *Mas-Herrero et al. (2014)* was the first to identify a group of healthy people for whom music is not rewarding. The term coined—specific musical anhedonia—refers to a unique subset of the population that draws no pleasure at all from music despite being perfectly able to experience pleasure in other ways. Using a stepwise regression analysis, Mas-Herrero et al. found the BMRQ score to be the only predictor of high-pleasure or chill responses in all their participants (compared to other reward scales such as the BIS/BAS). Their work has shown that the ability of music to induce pleasure may not be universal, and that there may be individual differences in access to the reward system (*Mas-Herrero et al., 2014*). To further understand the neural correlates behind musical pleasure and reward processing, further studies in that population are required and the BMRQ could prove to be a very valuable tool.

## CONCLUSION

The French version of the BMRQ appears valid and reliable. The addition of the French adaptation to the previously available English and Spanish versions significantly increases the reach of this scale. We believe it can not only serve as a valuable psychophysical tool in music reward and emotion research, but its use could also be extended to emotion and reward research in other domains and modalities in which music can be used to test the specificity of a given deficit. Clinical applications of the BMRQ include the examination of musical pleasure experience in healthy individuals and in patients suffering from a wide variety of cognitive, neurologic and auditory disorders.

The French BMRQ test is available online at the following URL: www.brainvitge.org/bmrq_french.php.

## ACKNOWLEDGEMENTS

We thank Anne Caclin, Lesly Fornoni and Yohana Lévêque from the Lyon research team on congenital amusia for their collaboration in this translation project into French as well as Gerard Mick for help in distributing the announcement. We also thank Robert Zatorre for his scientific input and help in coordinating this effort.

### Funding

The authors received no funding for this work.

### Competing Interests

The authors declare there are no competing interests.

### Author Contributions

- Joe Saliba and Alexandre Lehmann conceived and designed the experiments, performed the experiments, analyzed the data, contributed reagents/materials/analysis tools, wrote the paper, prepared figures and/or tables, reviewed drafts of the paper.
- Urbano Lorenzo-Seva analyzed the data, contributed reagents/materials/analysis tools, wrote the paper, prepared figures and/or tables, reviewed drafts of the paper.
- Josep Marco-Pallares conceived and designed the experiments, analyzed the data, contributed reagents/materials/analysis tools, reviewed drafts of the paper.
- Barbara Tillmann and Anthony Zeitouni conceived and designed the experiments, performed the experiments, contributed reagents/materials/analysis tools, reviewed drafts of the paper.

### Human Ethics

The following information was supplied relating to ethical approvals (i.e., approving body and any reference numbers):

1. McGill University, Faculty of Medicine Institutional Review Board
2. A11-E88-14B.

## Data Availability

   Data is available as Supplemental Information.

## Supplemental Information

Supplemental information for this article can be found online at http://dx.doi.org/10.7717/peerj.1760#supplemental-information.

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
