# Peer review of "French validation of the Barcelona Music Reward Questionnaire"

_PeerJ, doi:10.7717/peerj.1760_

## Round 0.1 · original submission · Major Revisions

Dear Authors,
There are fundamental statistical analysis (methodology) issues that need to be looked into. Please revise and redo analysis carefully accordingly to Peer Reviewer 1 on statistical methodology and on discussion and other aspects according to Peer Reviewers 2 and 3 .

·

Basic reporting

Leading zero
Please write as 0.88 instead of .88.
PLEASE CORRECT IN THE WHOLE REPORT.

L 137
"double translation"
It is commonly known as "forward and backward translation."
PLEASE USE THE COMMON TERM, PLEASE.

L 138
"two groups of bilingual (French/English) researchers"
Please write as :
two groups of bilingual (French and English) researchers
"/" stands for "OR". It is not appropriate.
PLEASE CORRECT IN THE WHOLE REPORT.

L138
Forward translation was mentioned as two groups of bilingual ....
PLEASE SPECIFY HOW MANY TRANSLATORS IN EACH GROUP.

L 159
When you were collecting the data online, how would you know that, the respondent is fluent in French??
HOW WAS IT CHECKED?
The same for age above 18. HOW WAS IT CHECKED?

L 194
"Mean age: 22.3 ± 7.8 years ...""
What is the number 7.8? It is not clear. If it is SD, please write as follows:
"Mean age was 22.3 (SD 7.8) years ...."
PLEASE USE THIS FORMAT FOR THE WHOLE REPORT.
NEVER USE THE SIGN "±"

L 221
by Mas-Herrero (0.22 – 0.46),
PLEASE WRITE AS "(0.22 to 0.46)"
Putting "-" in between two numbers is like a "negative sign"
PLEASE DO NOT DO.

THE HISTOGRAME "GLOBAL SENSITIVITY SCORE"
If score is given 1 to 5 for 20 items, it should be ranging from 20 to 100. The graph does not reflect this.
The graph has score less than 20.
PLEASE CLARIFY.

TABLE 1
REMOVE "±" AS MENTIONED ABOVE
REMOVE "-" AS MENTONED ABOVE
RANGE IS A SINGLE VALUE (MAXIIMUM MINUS MINIMUM). IF YOU WANT TO WRITE TWO VALUES, PLEASE CALL IT MINIMUM AND MAXIMUM RATHER THAN RANGE.

TABLE 2
IT IS UNNECESSARY TO PRESENT 95% CI.
IT IS WRONG TO PRESENT 95% CI IN THIS MANNER. 95% CI SHOULD BE PRESENTED WITH TWO VALUES AS LOWER LIMIT AND UPPER LIMIT. IT SHOULD NEVER BE PRESENTD WITH "±"
INSTEAD, PLEASE PRESENT WITH "MEAN (SD)"

TABLE 3
Some of the loadings are negative sign though they were grouped together with other positive items.
RESEARCHER SHOULD RECHECK THAT THESE "NEGATIVE ITEMS" SHOULD BE SCORE REVERSED BEFORE THE ANALYSIS.
PLEASE CONSULT STATISTICIAN PROPERLY. THIS IS UNACCEPTABLE TO PUBLISH AS IT DOESN'T HOLD THE VALIDITY.
PLEASE PUT LEADING ZERO IN THESE ALL DECIMAL VALUES.

END.

Experimental design

This study is not experimental.

Validity of the findings

Table 3 with negative signs on factor loadings do not show the good validity of the questionnaires. Reanalysis with reverse coding may be required.

Additional comments

1) It is a good important study.
2) Minor presentation related improvement need to be done.
3) Major reanalysis (maybe with reverse coding) is required. Negative signs in factor loadings that you include in the same group of facets are not valid to do so.

Reviewer 2 ·

Basic reporting

No comments

Experimental design

The expeirmental design is straight forward and appropriate

Validity of the findings

no comments

Additional comments

I find the paper written in a concise and appropriate manner for the most part. I found the general comments on the distribution of the French language not necessary and would recommend to drop them from the introduction.

Reviewer 3 ·

Basic reporting

ll 58 – 68: This section seems out of place, I understand that the mentioned neural pathways and structures are considered to be involved in reward experiences, but this is not made clear. The results are also not discussed in reference to these findings.
l 70: “inter-individual variability in music-induced reward” – could this be expanded? How does music-induced reward differ between people?
Introduction: There is no mention of the literature on anhedonia, which is discussed later on. I think it would be helpful for readers to be introduced to this topic in this section.
l 104: I would suggest adding a brief explanation about each facet and what it entails.
ll 176 – 186, 214 – 217: I would suggest rephrasing these sentences somewhat, as they can be found verbatim in Mas-Herrero et al., 2013.
l 213: I am a little confused where this “control scale” is coming from.
ll 363 – 365: This paper is listed twice in the reference section.
Figure 1: Is it possible to include a graph for the data from Mas-Herrero et al. (2013) for comparison?

Experimental design

No Comments

Validity of the findings

No Comments

Additional comments

Thank you for providing a French version of the BMRQ. I have some suggestions for editing revisions that would help to make this manuscript more readable in my opinion (see “Basic Reporting”). Best regards.

---

## Round 0.2 · Minor Revisions

Please proceed to do the minor revisions as suggested by the Reviewer 1, who is a statistician, so that it can be accepted in the next round of submission.

·

Basic reporting

With the following minor amendment, I recommend to publish the paper:

Figure 1:
As authors explained, the label of X axis should be "Global sensitivity to music reward scores (factor score)"
instead of "Global sensitivity to music reward scores"

Table 1
"N=1027" should be "n=1027"
Capital "N" refers to population size and small "n" refers to sample size.

L 226
95% CI has been removed from Table 2. Need to rewrite the sentence accordingly.

L 237
"AC" was not explained before using the acronym. Is it refering to "Acquiescence" in the Table 3? Please explain AC in the text.

Experimental design

No relevant

Validity of the findings

No relevant

Additional comments

No relevant

Reviewer 2 ·

Basic reporting

The authors have carefully responded to the comments of the three reviewers. I have no further questions or requests.

Experimental design

The authors have carefully responded to the comments of the three reviewers. I have no further questions or requests.

Validity of the findings

The authors have carefully responded to the comments of the three reviewers. I have no further questions or requests.

Additional comments

The authors have carefully responded to the comments of the three reviewers. I have no further questions or requests.

Reviewer 3 ·

Basic reporting

No Comments

Experimental design

No Comments

Validity of the findings

No Comments

Additional comments

Thank you for addressing my comments. Best regards.

---

## Round 0.3 · accepted · Accept

Dear Authors,Thank you for the submission of your revised manuscript which is now deemed suitable for publication in Peer J.

Reviewer 2 ·

Basic reporting

Everything fine -- I recommend acceptance

Experimental design

Everything fine -- I recommend acceptance

Validity of the findings

Everything fine -- I recommend acceptance

Additional comments

Everything fine -- I recommend acceptance